# Ameliorating the impairment of glucose utilization in a high-fat diet-induced obesity model through the consumption of Tucum-do-Cerrado (*Bactris Setosa Mart.*)

**Ananda de Mesquita Araújo**[1]*, **Sandra Fernandes Arruda**[2]

**1** Graduate Program in Human Nutrition, Faculty of Health Sciences, Campus Universitário Darcy Ribeiro, Universidade de Brasília, Brasília, Distrito Federal, Brazil, **2** Department of Nutrition, Faculty of Health Sciences, Campus Universitário Darcy Ribeiro, Universidade de Brasília, Brasília, Distrito Federal, Brazil

* nutriananda.araujo@gmail.com

**Data Availability Statement:** All relevant data are within the manuscript.

**Funding:** The authors received no specific funding for this work.

## Abstract

### Introduction

We evaluated the effect of Tucum-do-Cerrado on glucose metabolism homeostasis and its relationship with redox-inflammatory responses in a high-fat (HF) diet-induced obesity model.

### Results

The HF diet increased energy intake, feed efficiency, body weight, muscle and hepatic glycogen, insulin, homeostatic model assessment of insulin resistance (HOMA IR) and beta (β)-cell function, and gut catalase (CAT) activity, and decreased food intake, hepatic glutathione reductase (GR), glutathione peroxidase (GPX), glutathione S-transferase (GST), and superoxide dismutase (SOD) activities, hepatic phosphoenolpyruvate carboxykinase 1 (Pck1), and intestinal solute carrier family 5 member 1 (Slc5a1) mRNA levels compared with the control diet. However, the HF diet with Tucum-do-Cerrado decreased hepatic glycogen, and increased hepatic GR activity, hepatic Slc2a2 mRNA levels and serum Tnfa compared with the HF diet. Tucum-do-Cerrado decreased muscle glycogen, intestinal CAT and GPX activities, muscle PFK-1 and HK activities, and increased hepatic protein (CARB) and intestinal lipid (MDA) oxidation, hepatic GST activity, serum antioxidant potential, hepatic phosphofructokinase-1 (PFK-1) activity, intestinal solute carrier family 2 member 2 (Slc2a2), tumor necrosis factor (Tnf), interleukin-1 beta (Il1b), muscle protein kinase AMP-activated alpha 1 (Prkaa1), solute carrier family 2 member 2 (Slc2a2) mRNA levels, and serum interleukin-6 (IL-6) levels, regardless of diet type.

### Conclusion

Tucum-do-Cerrado consumption may ameliorate impaired glucose utilization in a HF diet-induced obesity model by increasing liver and muscle glucose uptake and oxidation. These data suggest that Tucum-do-Cerrado consumption improves muscle glucose oxidation in

**Competing interests:** The authors have declared that no competing interests exist.

non-obese and obese rats. This response may be related to the improvement in the total antioxidant capacity of rats.

## 1. Introduction

The prevalence of obesity and overweight population has doubled since 1980 [1]. According to the World Health Organization, 39% of adults aged 18 years and over were overweight in 2016, and 13% were obese [2]. Obesity is a metabolic disorder characterized by the excessive accumulation of adipose tissue associated with low-grade chronic inflammation and oxidative stress, which may impair health [3–5]. Chronic low-grade systemic inflammation and oxidative stress have been linked to the development of chronic diseases such as metabolic syndrome, type 2 diabetes, cardiovascular diseases, and some types of cancer [6, 7].

Therefore, effective strategies to prevent and treat obesity are necessary to reduce the burden of chronic diseases. Dietary components, such as polyphenols, found mainly in fruits and vegetables, have been associated with health promotion and, consequently, the prevention and reduction of obesity and metabolic complications associated with obesity [8]. Mezhibovsky et al. (2021) [9] showed that mice treated with a western diet (high in fats and sugars) supplemented with 1% grape polyphenols had higher lean mass, energy expenditure, and lower body weight. Grape polyphenols also enhanced ileal mRNA levels of glucose transporter-2 (Slc2a2, indicating higher glucose uptake) and the expression of ketohexokinase. The authors suggested that grape polyphenols attenuated diet-induced obesity and increased intestinal carbohydrate oxidation. Cinnamaldehyde (10mg/kg i.p.), a bioactive component of cinnamon, increased adipose tissue lipolysis, decreased fasting induced hyperphagia and inflammation in high fat diet-fed mice, suggesting its anti-obesity role [10], although these effects were not observed when a dose of 5mg/kg i.p. was used. Ballard et al. (2020) [11] showed that extracts of Tucum-do-Pantanal and Taruma-do-cerrado (100mg/kg), Brazilian fruits rich in polyphenols, prevented diet-induced body weight gain with a tendency to increase hepatic AMPK phosphorylation. However only the Tucum-do-Pantanal extract improved glucose metabolism by an improvement in the insulin tolerance test and a reduction in the insulin fasting level. Resveratrol, a polyphenol found in grapes and wine, inhibited α-glucosidase activity and decreased postprandial hyperglycemia in HF diet-fed mice (30 mg/kg body weight), indicating that a delay in the digestion and absorption of carbohydrates may suppress postprandial hyperglycemia [12].

Tucum-do-Cerrado (*Bactris setosa* Mart.) is a palm shrub fruit found in the Brazilian savanna and is characterized by a high content of polyphenols from different classes [13]. Flavanols, anthocyanins, and flavones are the major classes of phenolic compounds found in Tucum-do-Cerrado. Higher polyphenol content and antioxidant activity is found in Tucum-do-Cerrado peel than in the pulp [14]. Several *in vivo* studies conducted by our group have demonstrated the beneficial effects of Tucum-do-Cerrado on health. Heibel et al. (2018) [15] demonstrated that Tucum-do-Cerrado consumption induced hepatic Prkaa1α and Prkaa2 α, and consequently inhibited gluconeogenic rate limiting enzyme, glucose-6-phosphatase, and upregulated GLUT-2 glucose uptake. In an animal model of iron-induced oxidative stress, the consumption of Tucum-do-Cerrado protected tissues against oxidative damage by reducing iron availability in the liver and consequently inhibited Hamp expression (hepcidin) [16]. In another study, the authors observed that Tucum-do-Cerrado had an anti-aging effect, enhancing $NAD^+$-dependent histone/protein deacetylase sirtuin 1 (SIRT1) expression, which

activated the nuclear factor erythroid 2-related factor 2 (NRF2) pathway that attenuates oxidative damage to proteins and the inflammatory response induced by excess iron [17].

Considering the high polyphenol content and antioxidant and anti-inflammatory properties of Tucum-do-Cerrado, this study evaluated its' effect on the homeostasis of glucose metabolism and its interrelationships with redox-inflammatory responses in HF diet-induced obesity.

## 2. Methods

### 2.1. Fruits

The Tucum-do-Cerrado (Bactris setosa Mart.) fruits were purchased in Terezópolis de Goiás, Goiás, Brazil, in March, at full maturity. The fruits were washed with distilled water and stored at -80°C. After, in an environment protected from light, the seeds were removed and discarded from the frozen fruits, while the pulp and peel were homogenized using a mixer, freeze-dried (Beta 2–8 LSC plus, Martin Christ, Nova Analítica Ltda, São Paulo, Brazil) and stored at -80°C until rats' feed was prepared. The amount of Tucum-do-Cerrado fruit (150 g of pulp and peel fresh weight/kg of feed) added to rats' feed was defined according to our previous study [17], which showed that this amount of Tucum-do-Cerrado had an anti-aging effect. After the freeze-drying process, the 150 g of fresh Tucum-do-Cerrado pulp and peel yielded 28 g of powder. Therefore, 28 g of freeze-dried pulp and peel of Tucum-do-Cerrado/kg of diet were mixed with the other diet ingredients until a homogenous mixture was obtained, and then the diet was pelleted.

### 2.2. Treatment

The experiment with animals was conducted following the guidelines Animal Research: Reporting of In Vivo Experiments (ARRIVE). 24 newborn male Wistar rats (21 days), mean weight $67.4 \pm 6.0$g, were individually housed in stainless steel cages in a room with light/dark cycles of 12/12 h and temperature of $22 \pm 1$°C. The rats had free access to water and access to diet during the dark cycle.

After seven days of acclimatization with the standard rodent diet AIN-93G [18], the animals were randomly assigned to one of four experimental groups (n = 6): control group (CT-), which was fed with an AIN-93G diet; control + Tucum-do-Cerrado group (CT+) which was fed with an AIN-93G diet containing 28 g of freeze-dried Tucum-do-Cerrado pulp and peel/kg diet; high-fat group (HF-) which was fed with the AIN-93G diet containing 58% fat–(fat sources were 51.9% of lard and 6.1% of soybean oil, according to the Obesity Induction Diet described by the Research Diet [19] or high-fat + Tucum-do-Cerrado group (HF+) which was fed with an AIN-93G containing 58% fat and 28 g of freeze-dried Tucum-do-Cerrado pulp and peel/kg diet. After eight weeks of treatment, the rats were anesthetized using isoflurane, and the blood was collected by cardiac puncture. The liver, muscle and small intestine were excised, rinsed with saline, frozen in liquid nitrogen, and stored at -80°C until analysis. The daily food intake of the rats and their weekly weight measurements were recorded. The feed efficiency ratio was calculated according to the equation (Eq 1). The authors of the study were involved in overseeing the experiment and were aware of the group allocation to ensure proper implementation.

Eq 1 –Feed efficiency ratio

$$Feed\ efficiency = \frac{Body\ weight\ gain\ (g)}{Food\ intake\ (g)} \tag{1}$$

## 2.3. Glycemia and insulin

Fasting serum glucose concentration was determined using a glycosometer (Accu-check, Roche, Indianapolis, USA). Fasting plasma insulin concentration was determined by an enzyme-linked immunosorbent assay kit (ELISA) according to the manufacturer's assay protocol (Millipore Corporation, Missouri, USA). The indices insulin resistance (HOMA IR) and β-cell function (HOMA beta) were calculated using the below-mentioned Eqs 2 and 3.

Eq 2 –Indice insulin resistance (HOMA IR)

$$HOMA\ IR = \frac{blood\ glucose\ (mmol/L) x\ serum\ insulin\ (mU/L)}{22.5} \tag{2}$$

Eq 3 –Indice β-cell function (HOMA beta)

$$HOMA\ beta = \frac{20\ x\ insulinemia\ (U/L)}{blood\ glucose\ (mmol/L) - 3.5} \tag{3}$$

## 2.4. Hepatic and muscle glycogen concentration

Glycogen concentration in the liver and muscle was determined using the method described by Lo et al. (1970) [20]. A reaction mixture containing 50 mg of tissue and 500 μL of 30% potassium hydroxide solution saturated with sodium sulfate was incubated at 98°C for 30 min in a water bath. After cooling the sample on ice, a 95% ethanol solution (1.2 x vol) was added to the mixture, followed by a 30-min incubation on ice for glycogen precipitation. After sample centrifugation (840 x g for 30 min at 4°C), the glycogen precipitate was resuspended in 1 and 2 mL of distilled water for muscle and liver, respectively, and 200 mL of 5% phenol solution and 1 mL of 98% sulfuric acid solution were added to the reaction mixture. The reaction was incubated for 10 min at room temperature, followed by 20 min at 30°C in a water bath, and the absorbance was recorded at 490 nm (TCC-240A spectrophotometer, Shimadzu, Kyoto, Japan). A standard curve was constructed using purified type IV bovine glycogen standard solutions in the concentration range of 0–100 μg/mL. The glycogen concentration was expressed as mg glycogen/mg tissue.

## 2.5. Oxidative damage to lipids (malondialdehyde) and proteins (carbonyl)

Hepatic and small intestine malondialdehyde concentration (MDA—a product of lipid peroxidation) was determined by fluorescence detection of the MDA-thiobarbituric acid complex, using excitation wavelength at 532 nm and emission wavelength at 553 nm (SpectraMax M3 Multi-Mode Microplate Reader, Molecular Devices, 211 Sunnyvale, CA, USA) [21]. A standard curve was obtained from the product of the acid hydrolysis of 1,1,3,3 tetraethoxypropane acid (TEP; Sigma, St. Louis, MO, USA) over a concentration range of 0.0 to 5.05 nmol/mL. Malondialdehyde (MDA) concentration in tissues was expressed as nmol MDA/mg total protein.

The concentration of carbonyl protein in the liver and small intestine was evaluated as described by Richert et al. (2002) [22]. The absorbance of the dinitrophenylhydrazine-carbonyl complex was monitored at 376 nm (Spectrophotometer, Shimadzu—TCC 240A, Kyoto, Japan). Tissue carbonyl concentrations were expressed as nmol carbonyl/mg total protein using the extinction coefficient of 22,000 mM$^{-1}$cm$^{-1}$.

## 2.6. Total serum antioxidant capacity

Serum antioxidant capacity was determined using the Total Ferric-Reducing Ability of Plasma assay (The FRAP Assay) according to the protocol described by Benzie and Strain (1996) [23]. The FRAP reagent was prepared daily using 0.3 mol/L acetate buffer (pH 3.6), 10 mmol/L 2,4,6-triazinetripyridyl in 40 mmol/L hydrochloric acid, and an aqueous solution of 20 mmol/

L ferric chlorides in a proportion of 10:1:1, and incubated at 37˚C for 30 min. The reaction mixture was composed of 900 μL of FRAP reagent, 30 μL of a serum sample, and 90 μL of deionized water. After homogenization, the absorbance was recorded at 593 nm/4 min. A standard curve was constructed with $FeSO_4$ solutions over a concentration range of 0 to 2,000 μmol/L. The serum antioxidant capacity was expressed as μmol $FeSO_4$/L of serum.

## 2.7. Specific activity of antioxidant enzymes catalase (CAT), glutathione reductase (GR), glutathione peroxidase (GPX), glutathione-S-transferase (GST) and superoxide dismutase (SOD) in liver and small intestine

**2.7.1. Catalase (CAT).** Catalase activity (CAT, EC 1.11.1.6) was determined by monitoring the consumption of $H_2O_2$ at 240 nm (spectrophotometer, Shimadzu—TCC 240A, Kyoto, Japan) and the extinction coefficient of 0.0394 $mM^{-1}$ $cm^{-1}$ for $H_2O_2$ [24]. One unit of catalase is defined as the amount of enzyme required to decompose 1 mmol of $H_2O_2$ per minute.

**2.7.2. Glutathione peroxidase (GPx).** Glutathione peroxidase (GPx, EC 1.11.1.9) activity was evaluated using $H_2O_2$ as a substrate in an assay coupled with glutathione reductase-catalyzed oxidation of nicotinamide adenine dinucleotide phosphate (NADPH) at 340 nm (spectrophotometer, Shimadzu—TCC 240A, Kyoto, Japan), using an NADPH extinction coefficient of 6.22 $mM^{-1}$ $cm^{-1}$ [25]. One unit of glutathione peroxidase was defined as the amount of enzyme required to oxidize 1 nmol of nicotinamide adenine dinucleotide phosphate per minute.

**2.7.3. Glutathione reductase (GR).** Glutathione reductase (GR, EC 1.6.4.2) specific activity was determined by monitoring the consumption of NADPH at 340 nm for 20 s, as described by Joanisse and Storey [26]. The enzyme activity was calculated using the molar extinction coefficient of NADPH of 6.22 $mM^{-1}$ $cm^{-1}$ at 340 nm. One unit of glutathione reductase corresponds to the amount of enzyme required to oxidize 1 nmol NADPH / min.

**2.7.4. Glutathione S-transferase (GST).** The specific activity of Glutathione-S-transferase (GST, EC 2.5.1.18) was determined by the detection of the S-2,4-dinitrophenyl-glutathione complex at 340 nm [26]. Enzyme activity was quantified using the 9.6 $mM^{-1}$ $cm^{-1}$ extinction coefficient of the S-2,4-dinitrophenyl-glutathione complex. One unit of glutathione-s-transferase was defined as the amount of enzyme required to produce 1 nmol of product/min.

**2.7.5. Superoxide dismutase (SOD).** The determination of SOD activity was performed according to the protocol described by Mccord (2001) [27]. The assay consists of measuring the ability to inhibit the reduction of cytochrome C catalyzed by superoxide ion. The kinetics of the cytochrome c reduction reaction was monitored by reading the absorbance at 550 nm. One unit (U) of SOD corresponded to the amount of enzyme necessary to inhibit by 50% the reduction of cytochrome c. The results were expressed as U enzyme/mg of protein.

## 2.8. Enzymatic activity of α-glucosidase (GLY), glucokinase (GK), hexokinase (HK) and glucose-6-phosphatase (G6Pase)

To measure α-glucosidase activity, frozen intestinal samples were homogenized 1:10 (w/v) in 0.9% saline solution in an electric homogenizer at 4˚C, followed by centrifugation at 10,000 x g/20 min at 4˚C. The homogenates for the analysis of glucokinase (GK), hexokinase (HK), and phospho-fructokinase-1 (PFK1) activity were prepared from the frozen liver and muscle samples homogenized in a 30 mM KF, 4 mM EDTA, 15 mM 2-mercaptoethanol and 250 mM sucrose buffer solution (pH 7.5) at 4˚C, in a proportion of 1:5 (w/v). Subsequently, the homogenates were centrifuged at 1,000 x g at 4˚C for 10 min [28].

Glucose-6-phosphatase (G6Pase) and phosphoenolpyruvate carboxykinase (PEPCK) homogenates were prepared at 4˚C in a buffer solution containing 50 mM HEPES, 100 mM KCl, 2.5 mM dithiothreitol, 1 mM EDTA and 5 mM MgCl2 using an electric homogenizer, in

a proportion of 1:10 (w/v). The homogenates were centrifuged at 11,000 x g at 4°C for 30 min. The supernatant was transferred to another tube and centrifuged at 105,000 x g at 4°C for 60 min. The pellet was resuspended in a sucrose/EDTA (0.25 M/1 mM) solution in a proportion of 2:1 (weight of homogenized tissue/sucrose/EDTA volume) and used to determine the G6Pase activity, while the cytosolic fraction (supernatant) was used to measure the activity of the PEPCK enzyme.

**2.8.1. Intestinal α-glucosidase.**   The enzymatic activity of α-glucosidase (GLY, EC 3.2.1.8) was evaluated according to the method described by Gopal et al. (2017) [29], with minor modifications. The assay is based on the formation of p-nitrophenol from the 4-nitrophenyl β-D-glucopyranoside, which was detected at 340 nm (SpectraMax M3 Multi-Mode Micro plate Reader, Molecular Devices, Sunnyvale, California, USA). The results were expressed as nmol p-nitrophenol/min/mg protein.

**2.8.2. Hepatic Glucokinase and muscular hexokinase activity.**   The activities of glucokinase (GK, EC 2.7.1.2) and hexokinase (HK, EC 2.7.1.1) were determined according to the method proposed by Mosa et al. (2015) [30]. The reaction consists in monitoring the formation of NADPH, catalyzed by glucose-6-phosphate dehydrogenase in the presence of glucose and tissue homogenates, at 340 nm for 3 min (TCC-240A spectrophotometer, Shimadzu, Kyoto, Japan). Glucokinase activity was determined by the difference between the formation of NADPH in the presence of 100 mM and 5 mM glucose in the reaction medium. Muscle hexokinase activity was determined using the amount of NADPH formed in the presence of 5 mM glucose. The enzymatic activity was expressed as nmol of NADPH/min/mg of protein, using a molar extinction coefficient of 6.22 mM$^{-1}$/cm$^{-1}$.

**2.8.3. Glucose-6-phosphatase activity in the liver.**   The activity of glucose-6-phosphatase (G6Pase) in the liver was evaluated by monitoring the production of molybdate-phosphorus complex, which absorbs energy at a wavelength of 840 nm (SpectraMax M3 Multi-Mode Microplate Reader, Molecular Devices, Sunnyvale, California, USA), as described by Baginski and collaborators (1974) [31]. The results were expressed as nmol of inorganic phosphate/min/mg protein.

**2.8.4. Phosphoenolpyruvate carboxykinase (PEPCK) in the liver.**   Hepatic phosphoenolpyruvate carboxykinase (PEPCK) enzyme activity was determined as described by Hayanga et al. (2016) [32]. The cytosolic fraction of the homogenates was used to determine the enzymatic activity by the malate dehydrogenase-coupled assay. The rate of NADH oxidation was monitored at 340 nm for 3 min (SpectraMax M3 Multi-Mode Microplate Reader, Molecular Devices, Sunnyvale, California, USA). The rate of NADH oxidation was calculated using a molar extinction coefficient of 6.22 mM$^{-1}$/cm$^{-1}$, and the enzymatic activity was expressed as nmol of NADH/min/mg of protein.

**2.8.5. Phosphofructokinase-1 (PFK1) in the liver and muscle.**   The activity of the phosphofructokinase-1 (PFK1) enzyme was determined as described by Coelho et al. (2007) [28], using the triose phosphate isomerase and glycerol-3-phosphate dehydrogenase enzymatic-coupled assay, which detects the oxidation NADH at 340 nm during 20 min (SpectraMax M3 Multi-Mode Microplate Reader, Molecular Devices, Sunnyvale, California, USA). The results were expressed as the oxidation of NADH/min/mg of protein.

## 2.9. Determination of transcript levels of Ppck1 (PEPCK), Prkaa1 and Prkaa2 (AMPK subunits α1 and α2), Slc2a2 (GLUT2), Slc2a4 (GLUT4), Slc5a1 (SGLT1)

**2.9.1. Total RNA extraction and cDNA synthesis.**   The extraction of total tissue RNA was done using TRIzol™ reagent (Invitrogen Inc., Burlington, ON, Canada) as described by da

**Table 1. Primers sequences used for amplification of genes by real-time PCR assays and the GenBank accession numbers.**

| GENE | Primers sequences (5'- 3') | GenBank accession number |
|---|---|---|
| Actβ (β-Actin) | GTCGTACCACTGGCATTGTG<br>CTCTCAGCTGTGGTGGTGAA | NM_031144 |
| Il1b (IL-1β) | CACCTCTCAAGCAGAGCACAG<br>GGGTTCCATGGTGAAGTCAAC | NW_047658 |
| Gapdh (GAPDH) | TGCCCCCATGTTTGTGATG<br>GCTGACAATCTTGAGGGAGTTGT | NW_0476961 |
| Prkaa1 (AMPK-α1) | GAAGTCAAAGCCGACCCAAT<br>AGGGTTCTTCCTTCGCACAC | NM_019142 |
| Prkaa2 (AMPK-α2) | ATGATGAGGTGGTGGAGCAG<br>GTGAATGGTTCTCGGCTGTG | NM_023991 |
| Pck1 (PEPCK) | GCC TGT GGG AAA ACC AAC<br>CAC CCA CAC ATT CAA CTT TCC A | NM_198780 |
| Scl2a2 (GLUT2) | AAAGCCCCAGATACCTTTACCT<br>TGCCCCTTAGTCTTTTCAAGC | NM_012879 |
| Slc2a4 (GLUT4) | TTGCAGTGCCTGAGTCTTCTT<br>CCAGTCACTCGCTGCTGA | NM_012751.1 |
| Slc5a1 (SGLT1) | GAAGGGTGCATCGGAGAAG<br>CAATCAGCACGAGGATGAAC | NM_013033.2 |
| Tnf (TNF) | AAATGGGCTCCCTCTCATCAGTTC<br>GTCGTAGCAAACCACCAAGCAGA | X66539 |

Cunha et al. (2014) [33], with some modifications. The tissues were homogenized using a cell/tissue disruptor (L-beader 6, Cotia, São Paulo, Brazil). The liver was homogenized in tubes prefilled with 1 mm zirconium beads using 2 cycles of 3,670 rpm/30 s (between cycles the samples were left on ice for 30 s), while for muscle, tubes prefilled with 2.8 mm stainless steel beads were used, and 3 cycles of 3,670 rpm/45 s (between cycles the samples were left on ice for 45 s). The intestine was homogenized in tubes prefilled with 3 mm zirconium beads using 4 cycles of 3,470 rpm/40 s (between cycles the samples were left on ice for 40 s).

Complementary DNA (cDNA) was synthesized from total RNA using the high-capacity cDNA reverse transcription kit with RNase inhibitor (Applied Biosystems, Foster City, CA, USA). The transcript levels of intestinal solute carrier family 2 member 2 (Slc2a2/Glut2) and solute carrier family 5 member 1 (Slc5a1); hepatic solute carrier family 2 member 2 (Slc2a2/Glut2), phosphoenolpyruvate carboxykinase 1 (Pck1), protein kinase AMP-activated catalytic subunit alpha 1 (Prkaa1) and protein kinase AMP-activated catalytic subunit alpha 2 (Prkaa2); and muscle solute carrier family 2 member 4 (Slc2a4/Glut4), protein kinase AMP-activated catalytic subunit alpha 1 (Prkaa1) and protein kinase AMP-activated catalytic subunit alpha 2 (Prkaa2) were determined using real-time polymerase chain reaction (StepOne Real-Time PCR System, Applied Biosystems, Singapore). A qRT-PCR reaction was assembled using 5.0 μL Fast SYBR Green Master Mix 2× reagent, 2.0 μL of cDNA, and 0.2 μmol/L of each primer, in a final volume of 10 μL. The sequences of the forward (FW) and reverse (RW) primers used for real-time PCR reactions are described in Table 1. The specificity of each amplicon was evaluated by melting curve. The comparative CT method was used to quantify the abundance of target gene mRNA, and the results were presented as $2-\Delta\Delta CT$ [34]. All samples were assayed in triplicate and were normalized to the housekeeping gene β-actin (Actb) or glyceraldehyde 3-phosphate dehydrogenase (Gapdh).

## 2.10. Cytokine serum levels

Commercial enzyme immunoassay (ELISA) kits were used to assess serum levels of IL-6 (;—Thermo Fisher, Waltham, MA, USA), IL-1β and TNF-α (Bender, MedSystems, Vienna, Austria) IL-6, IL-1β, and TNF-α.

## 2.11. Statistical analysis

Values are presented as least square means and 95% confidence interval (n = 6). The effects of the diet, Tucum-do-Cerrado, and diet × Tucum-do-Cerrado interactions were analyzed by a two-way analysis of variance (ANOVA). Homogeneity of the variances between treatments was assumed, and after variance analysis, when interaction between the factors were significant (p < 0.05) means were compared using Tukey's test. The box plot method was used to remove outliers. All analyses were conducted using PROC GENMOD in the SAS/STAT® software (SAS OnDemand, SAS Institute Inc., Cary, NC, USA), and statistical significance was set at p < 0.05.

# 3. Results

## 3.1. Physiological and biochemical variables

Table 2 shows the physiological and biochemical characteristics of the rats after eight weeks (56 days) of treatment. The HF diet promoted a higher body weight gain from the third week of treatment compared to the control group (p = 0.0322; Fig 1). The rats treated with the HF diet showed a reduction in total food intake (1.18-fold; p <0.0001), increased energy intake (1.13-fold), and feed efficiency (1.05-fold) compared to the control group rats (p <0.0001, 0.035, and 0.007, respectively). The consumption of Tucum-do-Cerrado did not alter any of these variables when combined with a control or HF diet. No effect of diet or Tucum-do-Cerrado was obtained regarding fasting plasma glucose concentration. Regarding the muscle glycogen concentration, there was a marginal interaction between diet and Tucum-do-Cerrado (p = 0.063), where HF diet consumption increased muscle glycogen concentration in relation to the control diet (1.5-fold; p = 0.019), whereas Tucum-do-Cerrado promoted a reduction in muscle glycogen content both when combined with the control (CT(-) × CT(+):2.1-fold; p = 0.007) and HF diet (HF(-) × HF(+):2.9-fold; p <0.001). Compared with the control diet, HF diet consumption did not significantly increase glycogen concentration in the liver. However, the consumption of Tucum-do-Cerrado combined with HF diet decreased hepatic glycogen concentration compared with the HF diet (HF(-) × HF(+) 3.0-fold; p = 0.0016), as shown in Table 2. Consumption of the HF diet increased fasting plasma insulin concentration (p <0.001), insulin resistance index (HOMA-IR) (p <0.001), and beta cell functional capacity index (HOMA-B) (p = 0.041) compared with the control diet (Table 2). Tucum-do-Cerrado consumption did not affect insulin levels or its indices.

Values are means and 95% confidence interval (CI), n = 6/group. Means on the same line without a common capital letter [A, B] differ (*P* < 0.05) and means on the same column without a common lowercase letter [a,b] differ (*P* <0.05). Diet: control vs. high-fat diet; TUC: with vs. without addition of Tucum-do-Cerrado; Diet x TUC: interaction between diet type vs addition of Tucum-do-Cerrado. CT(-): rats fed control diet AIN-93G; CT(+): rats fed a control diet AIN-93G added of Tucum-do-Cerrado; HF(-): rats fed a high-fat diet; HF(+): rats fed a high-fat diet added of Tucum-do-Cerrado. HOMA IR: homeostatic model assessment–insulin resistance. HOMA β: homeostatic model assessment– β cell function. TUC: Tucum-do-Cerrado (*Bactris setosa* Mart.).

## 3.2. Oxidative damage to lipids and proteins

Rats treated with Tucum-do-Cerrado exhibited higher levels of carbonyl in the liver (p = 0.021) and MDA in the intestine (p = 0.018) than those not treated with Tucum-do-Cerrado (Table 3). There were no significant differences in hepatic MDA or intestinal carbonyl levels between the groups.

**Table 2. Physiological and biochemical characteristics of rats fed with control diet (CT-), control diet added of Tucum-do-Cerrado (CT+), high fat diet (HF-), or high diet added of Tucum-do-Cerrado (HF+).**

| | Tucum-do-Cerrado | | | Two-way ANOVA P values | | |
|---|---|---|---|---|---|---|
| Diet | (-) | (+) | Mean (95% CI) | Diet | TUC | Diet x TUC |
| **Physiological Variables** | | | | | | |
| **Total food Intake (g / 56 d)** | | | | <0.0001 | 0.981 | 0.109 |
| CT | 1,021.7 (957.3–1,086.1) | 1,074.8 (1,010.5–1,139.2) | 1,048.2 (1,002.7–1,093.8) [a] | | | |
| HF | 912.1 (847.7–976.4) | 857.4 (793.0–921.8) | 884.73 (839.2–930.2) [b] | | | |
| Mean | 966.9 (921.4–1,012.4) | 966.1 (920.6–1,011.6) | | | | |
| **Energy Intake (kcal / d)** | | | | <0.0001 | 0.783 | 0.092 |
| CT | 72.0 (67.0–77.1) | 75.8 (70.7–80.8) | 73.9 (70.3–77.5) [b] | | | |
| HF | 86.3 (81.2–91.3) | 81.1 (76.0–86.1) | 83.7 (80.1–87.2) [a] | | | |
| Mean | 79.4 (75.6–82.7) | 78.4 (74.9–82.0) | | | | |
| **Feed Efficiency (g / kcal)** | | | | 0.035 | 0.627 | 0.381 |
| CT | 3.7 (3.6–3.9) | 3.8 (3.6–3.9) | 3.7 (3.6–3.8) [b] | | | |
| HF | 4.0 (3.8–4.1) | 3.9 (3.7–4.0) | 3.9 (3.8–4.0) [a] | | | |
| Mean | 3.8 (3.7–3.9) | 3.8 (3.7–3.9) | | | | |
| **Biochemical variables** | | | | | | |
| **Glucose (mg / dL)** | | | | 0.163 | 0.300 | 0.075 |
| CT | 173.0 (160.2–185.8) | 154.1 (141.3–167.0) | 163.6 (154.5–172.6) | | | |
| HF | 170.3 (157.4–183.1) | 175.4 (162.6–188.3) | 172.9 (163.8–181.9) | | | |
| Mean | 171.6 (162.6–180.7) | 164.8 (155.7–173.9) | | | | |
| **Muscle glycogen (mg / g)** | | | | 0.038 | < 0.0001 | 0.063 |
| CT | 1.78 (1.37–2.19) [A, b] | 0.84 (0.43–1.25) [B] | 1.31 (1.02–1.60) | | | |
| HF | 2.63 (2.22–3.03) [A, a] | 0.89 (0.48–1.30) [B] | 1.76 (1.47–2.05) | | | |
| Mean | 2.20 (1.92–2.49) | 0.87 (0.58–1.15) | | | | |
| **Hepatic glycogen (mg / g)** | | | | 0.430 | 0.016 | 0.019 |
| CT | 7.80 (4.20–11.40) | 7.70 (4.10–11.30) | 7.75 (5.20–10.30) | | | |
| HF | 13.73 (10.13–17.33) [A] | 4.64 (1.30–7.98) [B] | 9.18 (6.73–11.6) | | | |
| Mean | 10.80 (8.2–13.3) | 6.17 (3.71–8.62) | | | | |
| **Insulin (ng / mL)** | | | | <0.001 | 0.964 | 0.369 |
| CT | 1.89 (0.91–2.88) | 1.46 (0.58–2.34) | 1.68 (1.02–2.34) [b] | | | |
| HF | 3.44 (2.56–4.32) | 3.83 (3.03–4.64) | 3.63 (3.04–4.23) [a] | | | |
| Mean | 2.67 (2.00–3.33) | 2.65 (2.05–3.24) | | | | |
| **HOMA IR** | | | | <0.001 | 0.865 | 0.219 |
| CT | 20.52 (9.93–31.11) | 13.57 (4.09–23.04) | 17.04 (9.94–24.15) [b] | | | |
| HF | 36.27 (26.80–45.75) | 41.57 (32.92–50.22) | 38.92 (32.51–45.33) [a] | | | |
| Mean | 28.40 (21.29–35.50) | 27.57 (21.15–33.98) | | | | |
| **HOMA β** | | | | 0.041 | 0.609 | 0.366 |
| CT | 226.8 (127.1–326.6) | 155.9 (56.1–255.6) | 191.3 (120.8–261.9) [b] | | | |
| HF | 288.4 (188.7–388.1) | 308.3 (217.2–399.3) | 298.3 (230.8–365.9) [a] | | | |
| Mean | 257.6 (187.1–328.1) | 232.1 (164.5–299.6) | | | | |

Values are means and 95% confidence interval (CI)., n = 6/group. Means on the same line without a common capital letter [A, B] differ ($P < 0.05$) and means on the same column without a common lowercase letter [a,b] differ ($P < 0.05$). Two-way ANOVA P values for the following comparisons: Diet: control vs. high-fat diet; TUC: with vs. without addition of Tucum-do-Cerrado; Diet x TUC: interaction between diet type vs addition of Tucum-do-Cerrado. CT(-): rats fed control diet AIN-93G; CT(+): rats fed a control diet AIN-93G added of Tucum-do-

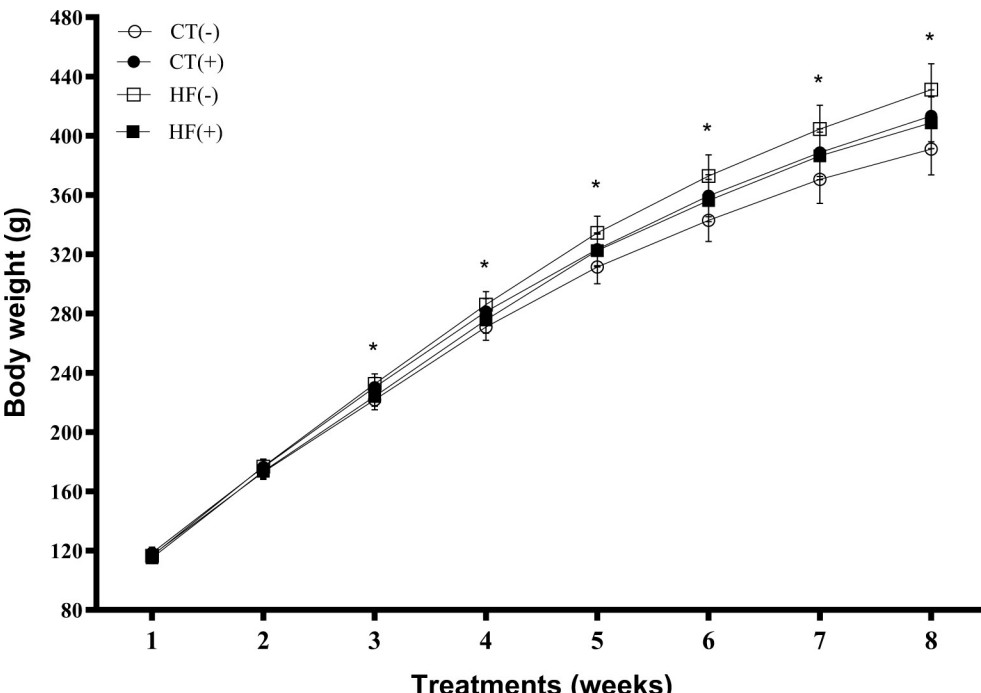

**Fig 1. Effect of high-fat diet and/or Tucum-do-Cerrado on body weight gain of rats during 8-weeks of treatment.** CT(-): rats fed control diet AIN-93G; CT(+): rats fed a control diet AIN-93G added of Tucum-do-Cerrado; HF(-): rats fed a high-fat diet; HF(+): rats fed a high-fat diet added of Tucum-do-Cerrado. Values are means and 95% confidence interval (CI), n = 6/group. *$p < 0.05$ Statistical differences for control vs. high-fat diet.

**Table 3. Effect of an 8-week treatment with high-fat diet and/or Tucum-do-Cerrado in hepatic and intestinal oxidative damage in rats.**

| | Tucum-do-Cerrado | | | *Two-way ANOVA P* values | | |
|---|---|---|---|---|---|---|
| Diet | (-) | (+) | Mean (95% CI) | Diet | Tucum | Diet x Tucum |
| | Hepatic oxidative damages (nmol / mg ptn) | | | | | |
| | MDA | | | | | |
| CT | 0.443 (0.343–0.544) | 0.362 (0.261–0.462) | 0.403 (0.332–0.474) | 0.510 | 0.667 | 0.252 |
| HF | 0.418 (0.317–0.518) | 0.455 (0.355–0.556) | 0.437 (0.366–0.508) | | | |
| Mean | 0.431 (0.360–0.502) | 0.409 (0.338–0.480) | | | | |
| | Carbonyl | | | | | |
| CT | 0.388 (0.303–0.474) | 0.514 (0.429–0.600) | 0.451 (0.391–0.512) | 0.351 | 0.021 | 0.645 |
| HF | 0.367 (0.282–0.453) | 0.453 (0.368–0.539) | 0.410 (0.350–0.471) | | | |
| Mean | 0.378 (0.318–0.438) [B] | 0.484 (0.424–0.544) [A] | | | | |
| | Intestinal oxidative damages (nmol/mg ptn) | | | | | |
| | MDA | | | | | |
| CT | 0.416 (0.282–0.549) | 0.694 (0.561–0.828) | 0.555 (0.460–0.650) | 0.470 | 0.018 | 0.124 |
| HF | 0.473 (0.340–0.607) | 0.537 (0.404–0.671) | 0.505 (0.411–0.600) | | | |
| Mean | 0.445 (0.350–0.539) [B] | 0.616 (0.521–0.710) [A] | | | | |
| | Carbonyl | | | | | |
| CT | 0.406 (0.304–0.509) | 0.441 (0.339–0.544) | 0.424 (0.351–0.497) | 0.766 | 0.145 | 0.414 |
| HF | 0.379 (0.276–0.482) | 0.500 (0.397–0.603) | 0.440 (0.367–0.512) | | | |
| Mean | 0.393 (0.320–0.465) | 0.471 (0.398–0.543) | | | | |

Cerrado; HF(-): rats fed a high-fat diet; HF(+): rats fed a high-fat diet added of Tucum-do-Cerrado.

## 3.3. Specific activity of antioxidant enzymes

In the liver, we observed an effect of the interaction between diet × Tucum-do-Cerrado on GR activity; the HF diet promoted a decrease in GR activity compared with the control diet (p <0.0001), whereas the consumption of Tucum-do-Cerrado combined with a HF diet increased GR activity compared with HF diet (p = 0.019; Fig 2A). Hepatic GPX and GST activity were lower in rats treated with HF diet rather than the control diet (p = 0.012 and 0.0016, respectively), while rats fed Tucum-do-Cerrado showed an increase in GST activity compared with the untreated group (p = 0.013). SOD activity in the liver was lower in rats treated with HF diet than in rats in the control group (p = 0.032), and the consumption of Tucum-do-Cerrado combined with the control diet decreased SOD activity compared with the control diet (p = 0.020).

In contrast to observations in the liver, the HF diet consumption increased CAT activity in the small intestine compared with the control diet (p = 0.034; Fig 2B), whereas the consumption of Tucum-do-Cerrado marginally decreased CAT activity (p = 0.062). A similar profile

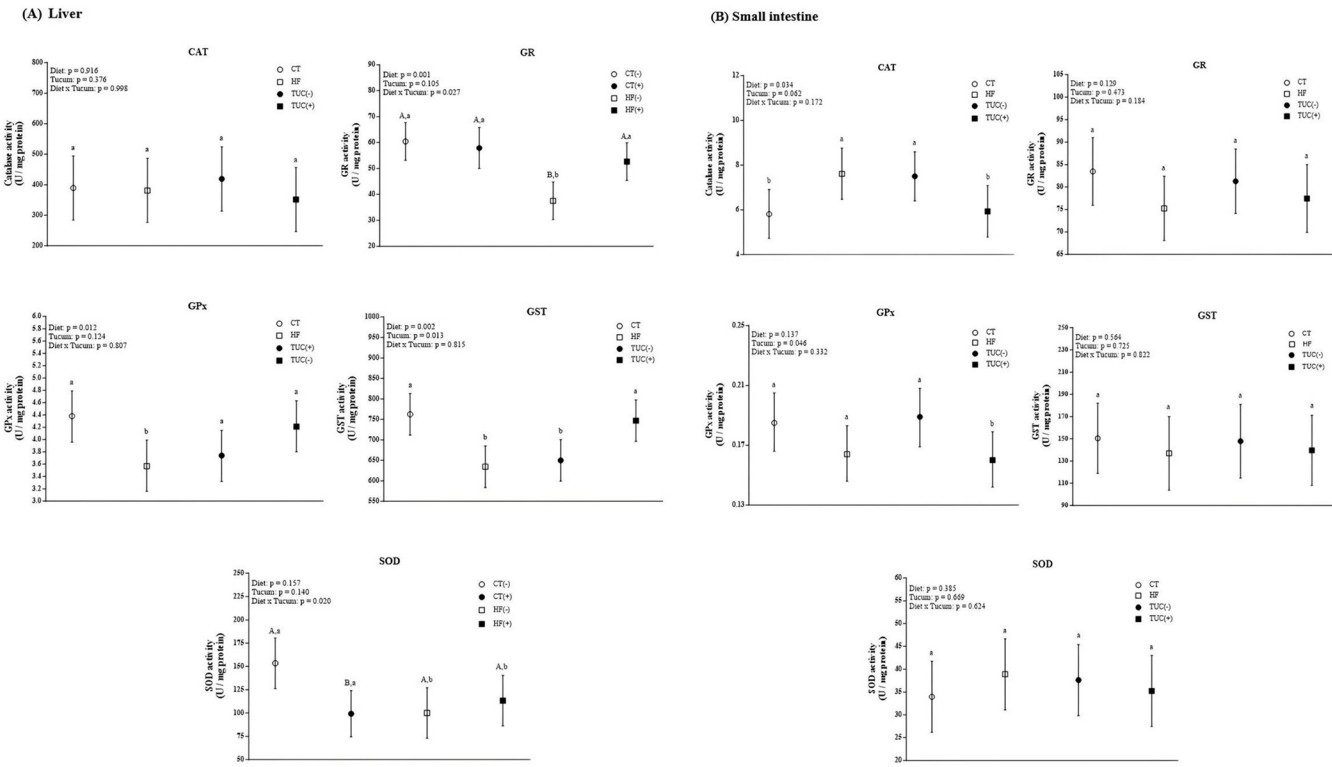

**Fig 2.** Effect of high-fat diet and/or Tucum-do-Cerrado consumption in antioxidant enzymes activity in the liver (A) and small intestine (B) of rats. Values are means and 95% confidence interval (CI), n = 6/group. For diet or Tucum-do-Cerrado effect: within the diet type and within the Tucum-do-Cerrado addition (+) or not (-), data without a common letter differ ($P < 0.05$). For diet x Tucum-do-Cerrado interaction: Different lowercase letters[a,b] indicate differences for control vs. high-fat within Tucum-do-Cerrado addition (+) or not (-); and different capital letters[A,B] indicate differences for addition (+) vs. not addition (-) of Tucum-do-Cerrado within control or high-fat diets ($P < 0.05$). CT: rats fed control diet AIN-93G; HF: rats fed a high-fat diet; TUC(-): rats not treated with Tucum-do-Cerrado; TUC(+): rats treated with Tucum-do-Cerrado. CT(-): rats fed control diet AIN-93G; CT(+): rats fed a control diet AIN-93G added of Tucum-do-Cerrado; HF(-): rats fed a high-fat diet; HF(+): rats fed a high-fat diet added of Tucum-do-Cerrado. CAT = catalase; GR = glutathione reductase; GPX = Glutathione peroxidase; SOD = superoxide dismutase; GST = glutathione-s-transferase. Two-way ANOVA *P* values for the following comparisons: Diet: control vs. high-fat diet; Tucum: with vs. without addition of Tucum-do-Cerrado independent of diet type; Diet x Tucum: interaction between diet type vs addition of Tucum-do-Cerrado.

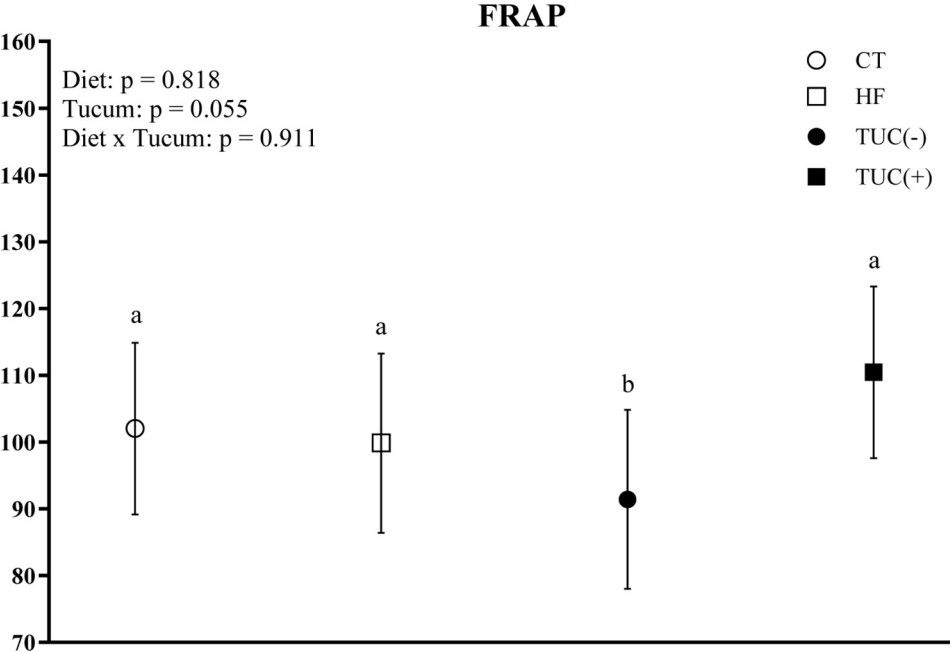

**Fig 3. Effect of high-fat diet and/or Tucum-do-Cerrado consumption in total serum antioxidant capacity of rats.**
Values are means and 95% confidence interval (CI), n = 5/group. For diet or Tucum-do-Cerrado effect: within the diet type and within the Tucum-do-Cerrado addition (+) or not (-), data without a common letter differ ($P < 0.05$). For diet x Tucum-do-Cerrado interaction: Different lowercase letters[a,b] indicate differences for control vs. high-fat within Tucum-do-Cerrado addition (+) or not (-) ($P < 0.05$). CT: rats fed control diet AIN-93G; HF: rats fed a high-fat diet; TUC(-): rats not treated with Tucum-do-Cerrado; TUC(+): rats treated with Tucum-do-Cerrado. FRAP = Ferric reducing antioxidant power. Two-way ANOVA *P* values for the following comparisons: Diet: control vs. high-fat diet; Tucum: with vs. without addition of Tucum-do-Cerrado independent of diet type; Diet x Tucum: interaction between diet type vs addition of Tucum-do-Cerrado.

was observed for GPX; the consumption of Tucum-do-Cerrado decreased GPX activity in the small intestine (p = 0.046). The activities of the antioxidant enzymes GR, GST, and SOD in the small intestine did not differ between diet type and Tucum-do-Cerrado consumption (Fig 2B). Regarding the total serum antioxidant capacity, the consumption of Tucum-do-Cerrado increased FRAP values in the serum, regardless of diet type (p = 0.055; Fig 3).

### 3.4. Carbohydrate metabolism markers in the liver

The consumption of a HF diet promoted a decrease in Pck1 mRNA levels in the liver, compared with the control diet (p <0.001). An effect of diet × Tucum-do-Cerrado interaction on hepatic Slc2a2 mRNA levels was observed (p = 0.043). Similarly, the consumption of Tucum-do-Cerrado with a HF diet promoted an increase in Slc2a2 mRNA levels in the liver compared to the HF diet (p = 0.055). However, the consumption of Tucum-do-Cerrado did not alter the hepatic mRNA levels of Prkaa1 or Prkaa2 (Fig 4A).

With regards to the activity of glycolytic enzymes in the liver, the consumption of Tucum-do-Cerrado increased PFK1 activity (p = 0.0062) regardless of diet type. No effects of diet and/or Tucum-do-Cerrado consumption on hepatic G6Pase were observed (Fig 4B).

### 3.5. Carbohydrate metabolism markers in the small intestine

Consumption of a HF diet promoted a decrease in Slc5a1 mRNA levels in the small intestine compared with the control diet (p = 0.041; Fig 5A), while no effect of Tucum-do-Cerrado on

**(A) mRNA levels**

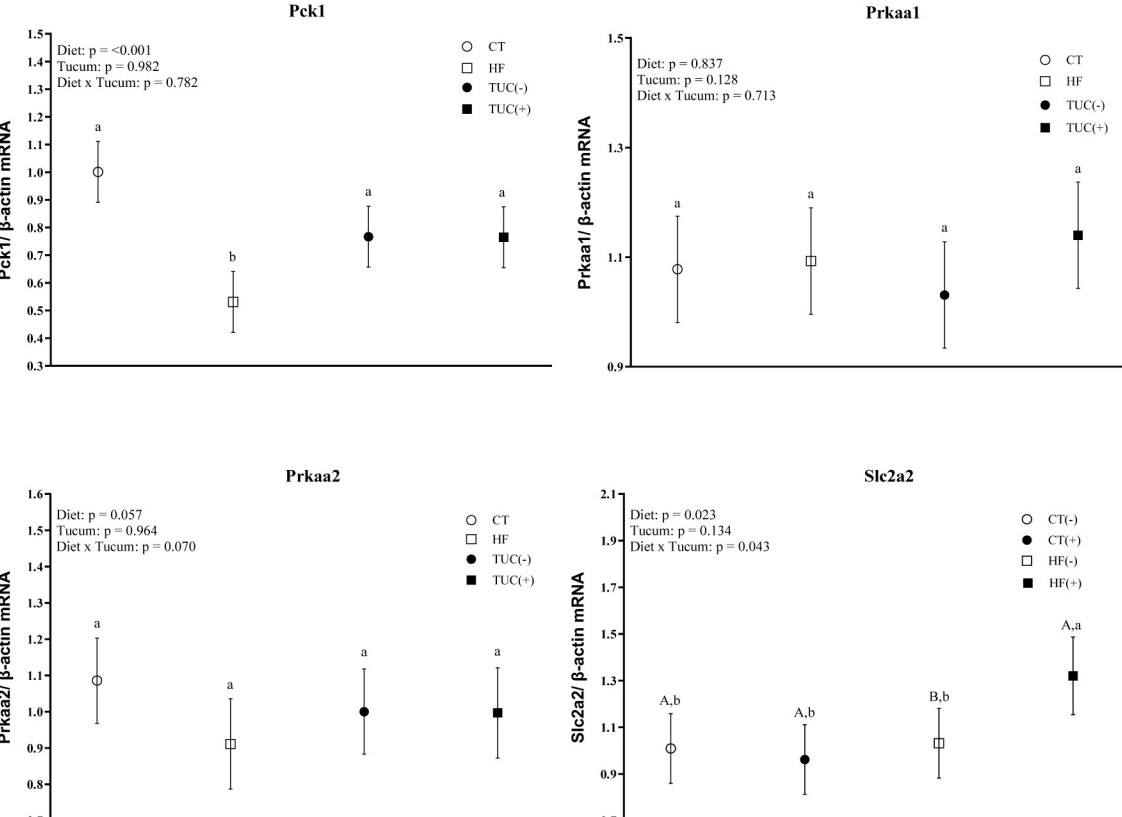

**(B) Enzymatic activity**

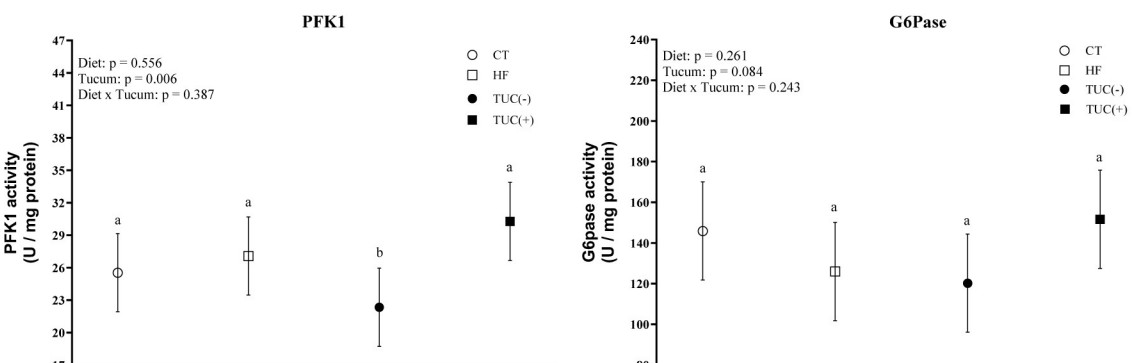

**Fig 4.** The mRNA levels of Pck1 (PEPCK), Prkaa1 (AMPK-α1), Prkaa2 (AMPK-α2) and Slc2a2 (GLUT2) (A) and Phosphofructokinase 1 (PFK1) and Glucose-6-phosphatase (G6Pase) specific activity (B) in the liver of rats treated for 8-weeks with one of the following diets: CT: rats fed control diet AIN-93G; HF: rats fed a high-fat diet; TUC(-): rats not treated with Tucum-do-Cerrado; TUC(+): rats treated with Tucum-do-Cerrado. CT(-): rats fed control diet AIN-93G; CT(+): rats fed a control diet AIN-93G added of Tucum-do-Cerrado; HF(-): rats fed a high-fat diet; HF(+): rats fed a high-fat diet added of Tucum-do-Cerrado. Values are means and 95% confidence interval (CI), n = 5/group. For diet or Tucum-do-Cerrado effect: within the diet type and within the Tucum-do-Cerrado addition (+) or not (-), data without a common letter differ ($P < 0.05$). For diet x Tucum-do-Cerrado interaction: Different lowercase letters[a,b] indicate differences for control vs. high-fat within Tucum-do-Cerrado addition (+) or not (-); and different capital letters[A,B] indicate differences for addition (+) vs. not addition (-) of Tucum-do-Cerrado within control or high-fat diets ($P < 0.05$). Two-way ANOVA *P* values for the following comparisons: Diet: control vs. high-fat diet; Tucum: with vs. without addition of Tucum-do-Cerrado independent of diet type; Diet x Tucum: interaction between diet type vs addition of Tucum-do-Cerrado.

**(A) mRNA levels**

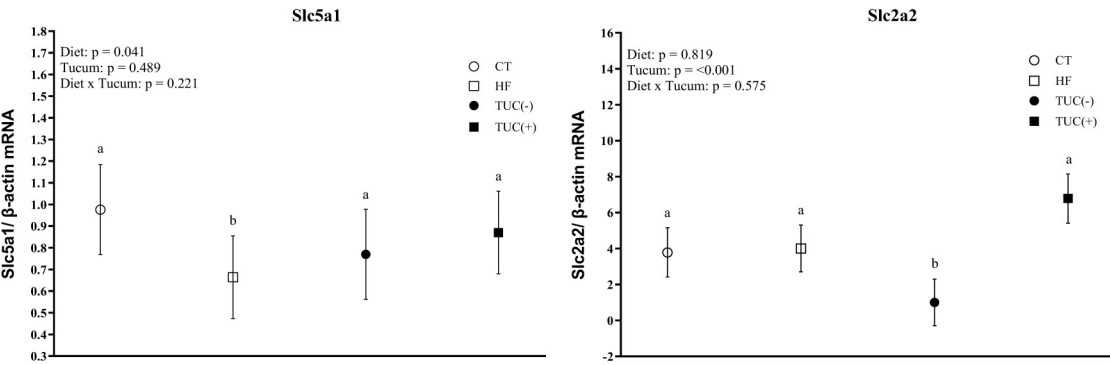

**(B) Enzymatic activity**

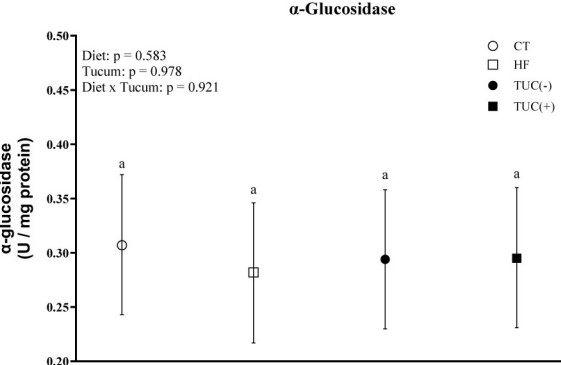

**Fig 5.** The mRNA levels of Slc5a1 (SGLT1) and Slc2a2 (GLUT2) in the small intestine (A) and α-Glucosidase specific activity in the small intestine (B), of rats treated for 8-weeks with one of the following diets: CT: rats fed control diet AIN-93G; HF: rats fed a high-fat diet; TUC(-): rats not treated with Tucum-do-Cerrado; TUC(+): rats treated with Tucum-do-Cerrado. Values are means and 95% confidence interval (CI), n = 5/group. For diet or Tucum-do-Cerrado effect: within the diet type and within the Tucum-do-Cerrado addition (+) or not (-), data without a common letter differ ($P < 0.05$). Two-way ANOVA *P* values for the following comparisons: Diet: control vs. high-fat diet; Tucum: with vs. without addition of Tucum-do-Cerrado independent of diet type; Diet x Tucum: interaction between diet type vs addition of Tucum-do-Cerrado.

Slc5a1 mRNA levels was observed. Tucum-do-Cerrado consumption increased the intestinal levels of Slc2a2 mRNA, regardless of the type of diet (p <0.001).

In relation to the activity of glycolytic enzymes in the small intestine, no effect of diet and/or Tucum-do-Cerrado consumption was observed in intestinal α-glucosidase activity (Fig 5B).

### 3.6. Carbohydrate metabolism markers in the muscle

Consumption of Tucum-do-Cerrado by rats increased the mRNA levels of Prkaa1 (p <0.0001) and Slc2a4 (p <0.0001) in the muscle, independent of diet type. However, there was no significant effect on Prkaa2 mRNA levels in the muscle of the rats (Fig 6A). Considering the activity of the rate-limiting enzymes of the glycolytic pathway, treatment of rats with Tucum-do-Cerrado decreased HK and PFK1 activity in the muscle, independent of the diet type (p <0.001 and 0.0215, respectively; Fig 6B).

**(A) mRNA levels**

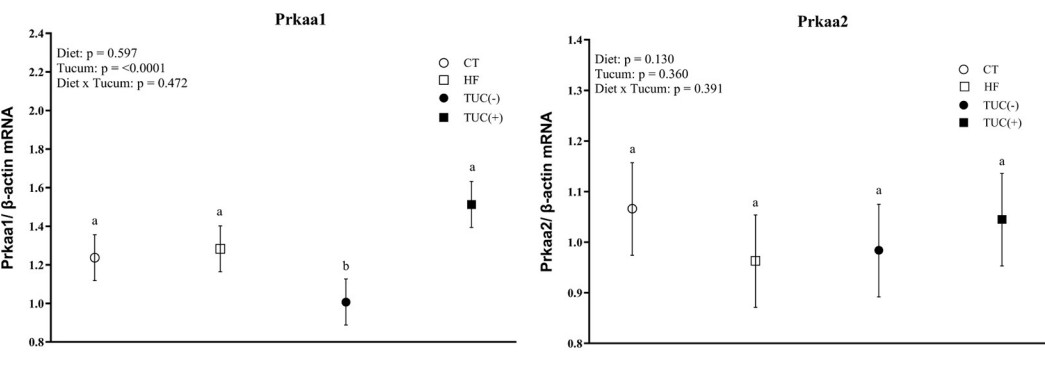

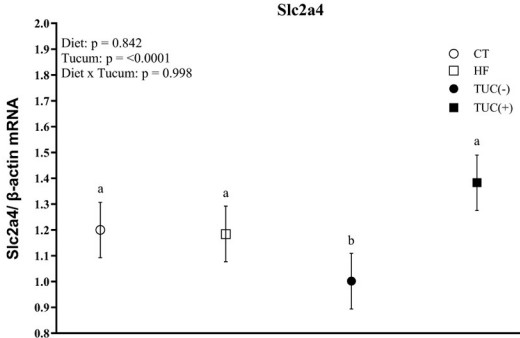

**(B) Enzymatic activity**

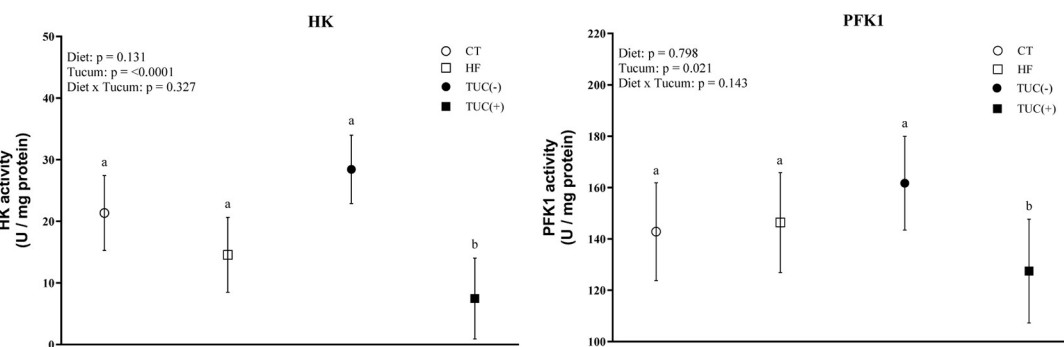

**Fig 6.** The mRNA levels of Prkaa1 (AMPK-α1), Prkaa2 (AMPK-α2) and Slc2a2 (GLUT2) (A) and hexokinase (HK) and phosphofructokinase (PFK1) specific activity (B) in the muscle of rats treated with one of the following diets: CT: rats fed control diet AIN-93G; HF: rats fed a high-fat diet; TUC(-): rats not treated with Tucum-do-Cerrado; TUC(+): rats treated with Tucum-do-Cerrado. Values are means and 95% confidence interval (CI), n = 5/group. For diet or Tucum-do-Cerrado effect: within the diet type and within the Tucum-do-Cerrado addition (+) or not (-), data without a common letter differ ($P < 0.05$). Two-way ANOVA *P* values for the following comparisons: Diet: control vs. high-fat diet; Tucum: with vs. without addition of Tucum-do-Cerrado independent of diet type; Diet x Tucum: interaction between diet type vs addition of Tucum-do-Cerrado.

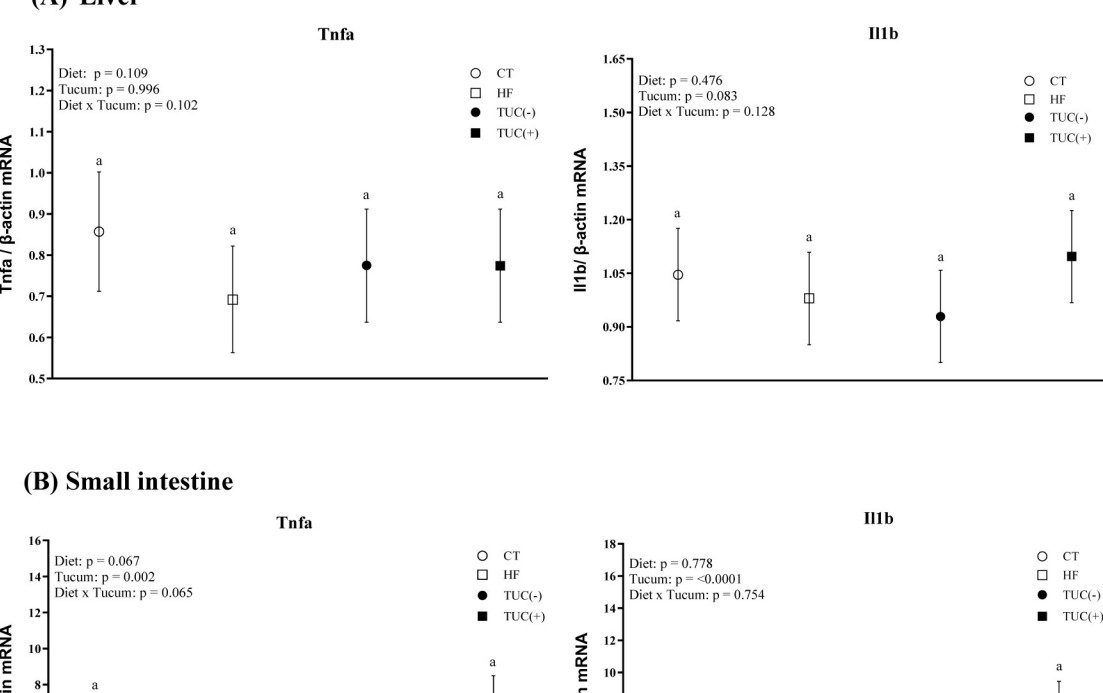

**Fig 7.** The mRNA levels of tumor necrosis alpha (Tnfa) and interleukin 1b (Il1b) in the liver (A) and small intestine (B) of rats treated with one of the following diets: CT: rats fed control diet AIN-93G; HF: rats fed a high-fat diet; TUC(-): rats not treated with Tucum-do-Cerrado; TUC(+): rats treated with Tucum-do-Cerrado. Values are means and 95% confidence interval (CI), n = 5/ group. For diet or Tucum-do-Cerrado effect: within the diet type and within the Tucum-do-Cerrado addition (+) or not (-), data without a common letter differ ($P < 0.05$). Two-way ANOVA $P$ values for the following comparisons: Diet: control vs. high-fat diet; Tucum: with vs. without addition of Tucum-do-Cerrado independent of diet type; Diet x Tucum: interaction between diet type vs addition of Tucum-do-Cerrado.

### 3.7. Inflammatory markers

Tnfa and Il1b mRNA levels were upregulated in the small intestine of groups treated with Tucum-do-Cerrado (p = 0.0018 and <0.001, respectively; Fig 7), regardless of the diet type. In the liver, no significant effects of diet and/or Tucum-do-Cerrado on Tnfa and Il1b mRNA levels were observed.

As shown in Fig 8, there was an interaction between diet and Tucum-do-Cerrado in terms of serum pro-inflammatory cytokine TNF-α concentration. The HF diet promoted a reduction in the serum TNF-α level compared to control diet (p = 0.013), while the addition of Tucum-do-Cerrado to the HF diet increased TNF-α serum levels in comparison to the HF diet (p <0.0001). Consumption of Tucum-do-Cerrado increased IL-6 serum levels independent of diet type (p <0.0001).

## 4. Discussion

This study shows the effect of Tucum-do-Cerrado, a Brazilian savanna fruit with high polyphenol [14] content, on glucose homeostasis and its interrelationships with redox and

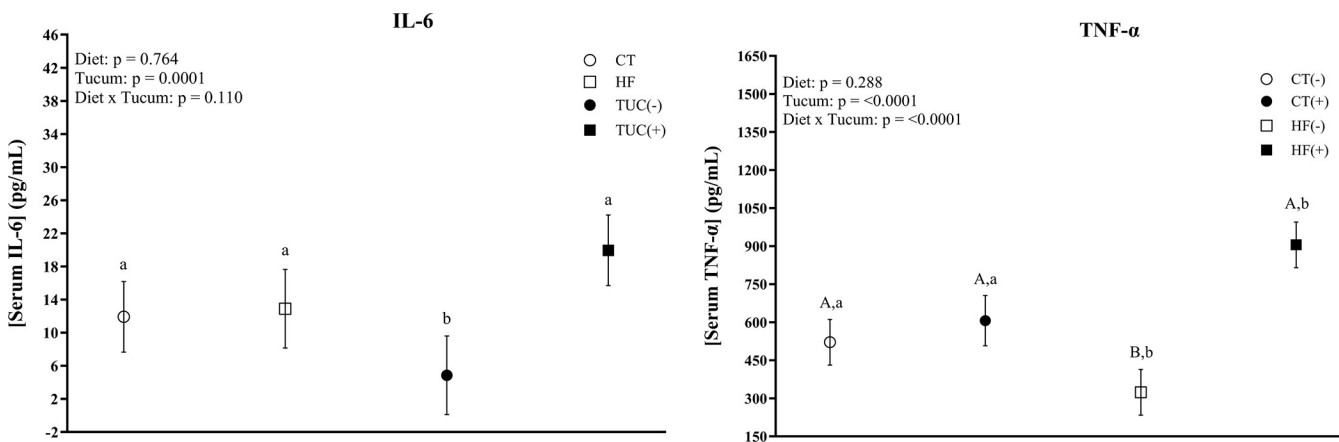

**Fig 8. Tumor necrosis alpha and interleukin-6 (IL-6) serum protein levels of rats fed with one of the following diets.** CT: rats fed control diet AIN-93G; HF: rats fed a high-fat diet; TUC(-): rats not treated with Tucum-do-Cerrado; TUC(+): rats treated with Tucum-do-Cerrado. CT(-): rats fed control diet AIN-93G; CT(+): rats fed a control diet AIN-93G added of Tucum-do-Cerrado; HF(-): rats fed a high-fat diet; HF(+): rats fed a high-fat diet added of Tucum-do-Cerrado. Values are means and 95% confidence interval (CI), n = 5/group. For diet or Tucum-do-Cerrado effect: within the diet type and within the Tucum-do-Cerrado addition (+) or not (-), data without a common letter differ (*P* < 0.05). For diet x Tucum-do-Cerrado interaction: Different lowercase letters[a,b] indicate differences for control vs. high-fat within Tucum-do-Cerrado addition (+) or not (-); and different capital letters[A,B] indicate differences for addition (+) vs. not addition (-) of Tucum-do-Cerrado within control or high-fat diets Two-way ANOVA *P* values for the following comparisons: Diet: control vs. high-fat diet; Tucum: with vs. without addition of Tucum-do-Cerrado independent of diet type; Diet x Tucum: interaction between diet type vs addition of Tucum-do-Cerrado. (*P < 0.05*).

inflammatory responses in an HF diet-induced obesity model. As expected, the HF diet promoted greater body weight gain as well as an increase in serum insulin concentration and glucose homeostatic indexes, even though blood glucose concentration was like that of the control diet, suggesting a metabolic impairment of glucose homeostasis. The higher glycogen concentration in the muscle and liver of rats treated with HF diet reinforces the above hypothesis, as the condition identified as "metabolic inflexibility" is characterized by an impairment of glucose oxidation due to high free fatty acids' availability and triglyceride accumulation due to the HF diet consumption [35, 36].

In contrast to our results, Ballard et al. (2020) [11] observed that the extracts of Tucum-do-Pantanal (*Bactris setosa* Mart.) and Taruma-do-Cerrado (*Vitex cymosa Bertero ex Spreng*), both Brazilian fruit sources of phytochemicals, decreased the body weight of diet-induced obese C57BL/6J mice after five weeks and in the last week of treatment, respectively. However, Mezhibovsky et al. (2021) [9] observed lower body weight gain in mice treated with diet-induced obesity supplemented with grape polyphenols, only from 15-week of treatment. Khare et al. (2016) [10] showed that Cinnamaldehyde 10mg/kg i.p. (a bioactive component of cinnamon) co-administered with a HF diet prevented body weight gain, whereas at a dose of 5mg/kg i.p., it did not, suggesting that the compound concentration may justify different results. In our study, rats fed a HF diet had reduced food intake by 11% compared to control rats, which attenuated body weight gain and resulted in moderate obesity (21% of body weight gain). These results suggest that some animals are resistant to HF diet-induced obesity [37] and may explain the non-effect of Tucum-do-Cerrado consumption on body weight gain during the 8-week treatment period. Furthermore, the higher concentration and bioavailability of polyphenolic compounds in extracts compared to fresh fruit, the route of administration (gavage *versus* mixed with diet), as well as the treatment time, obesity treatment model instead of prevention, and adult animals instead of newly weaned animals may be associated with the contradictory results obtained between studies [37–40].

Tucum-do-Cerrado consumption (150g fresh weight/kg diet) did not attenuate the detrimental effect on blood markers of glucose homeostasis compared to the HF(-) group. However, it decreased glycogen concentration in the liver and muscle when combined with a HF diet, regardless of the diet type. The lower hepatic glycogen concentration promoted by the consumption of Tucum-do-Cerrado when combined with a HF diet was associated with increased activity of the glycolytic enzyme phosphofructokinase-1 (PFK-1), suggesting that Tucum-do-Cerrado activated glycogenolysis, and consequently, the glycolytic pathway in a HF diet-induced obesity model.

In agreement with the above hypothesis, consumption of Tucum-do-Cerrado upregulated Slc2a2 mRNA levels (Glut-2) in the intestine independent of diet type and in the liver when combined with a HF diet. Therefore, we hypothesized that Tucum-do-Cerrado improves intestinal glucose uptake to compensate for the higher oxidation levels of glucose in hepatic cells and maintain blood glucose levels within the normal range. Similar to the findings of our study, Mezhibovsky et al. (2021) [9] showed that mice treated with diet-induced obesity (low in fiber, but high in fats and sugars) supplemented with grape polyphenols had higher ileal mRNA levels of glucose transporter 2 (Sl2a2) than those not supplemented. They suggested that grape polyphenols increase carbohydrate utilization by increasing intestinal glucose uptake and oxidation as a replacement for energy sources, since these mice had a lower concentration of butyrate (short-chain fatty acid), a major energy substrate of the distal intestine, and higher gene expression of duodenal pyruvate dehydrogenase (PDH). Therefore, we hypothesized that Tucum-do-Cerrado may have an effect similar to that observed for grape polyphenols, as no difference was observed in serum insulin concentration and insulin sensitivity as estimated by the HOMA-IR and HOMA-β indices.

Divergent responses regarding the glycolytic pathway in the liver and muscles were expected because of the opposing sets of these two tissues [41]. Glycolytic pathway is stimulated by the activation of glycolytic enzymes such as PFK1 and hexokinase (HK) [42, 43]. Fructose 2,6-bisphosphate allosterically activates PFK-1 more potently than its own product (fructose 1,6- bisphosphate). Considering that in the liver the activities of the bifunctional enzyme phosphofructokinase-2 (PFK-2)/ fructose 2,6-bisphosphatase (FBPase-2) are influenced by phosphorylation/dephosphorylation [43], we hypothesized that phytochemical compounds of Tucum-do-Cerrado lead to dephosphorylation of the PFK-2/FBPase-2 bifunctional enzyme and consequent activation of PFK-2, promoting an increase in the concentration of fructose 2,6-bisphosphate, which allosterically activates PFK-1 favoring glucose oxidation through glycolytic pathway.

AMP-activated protein kinase (AMPK) is a metabolic energy sensor that upregulates catabolic pathways that generate ATP and down-regulates anabolic pathways. Activation of AMPK increases GLUT-4 expression or translocation through an insulin independent mechanism [44]. Therefore, the upregulation of Prkaa1 (AMPK) and Slc2a4 mRNA (GLUT-4) levels, along with a decrease in muscle glycogen concentration in rats treated with Tucum-do-Cerrado, regardless of diet type, suggest an increase in muscle glucose uptake and oxidation. Unexpectedly, the activities of the enzymes of the glycolytic pathway, HK and PFK-1, decreased with Tucum-do-Cerrado consumption, regardless of diet type.

Owing to opposing metabolic sets in the liver and muscle tissues, divergent responses are expected regarding the regulation of the glycolytic pathway [41]. In the muscle, the increase in fructose 2,6-bisphosphate occurs because of the accumulation of hexose monophosphates produced from the cAMP-dependent activation of glycogenolysis [41, 45]. Therefore, in the present study, the lower glycogen concentration in the muscle of rats treated with Tucum-do-Cerrado may have decreased hexose monophosphate concentrations, inactivated PFK-2, and consequently decreased fructose 2,6-bisphosphate production by inhibiting glycolysis. Taken

together, these results suggest that Tucum-do-Cerrado stimulates glucose uptake and oxidation.

In addition, a previous study observed that Tucum-do-Cerrado increased hepatic nuclear factor erythroid 2-related factor 2 (Nfe2l2) mRNA and protein (Nrf2) levels [17]. Therefore, we also hypothesized that the dietary consumption of Tucum-do-Cerrado could induce Nrf2 expression and consequently stimulate glucose uptake and utilization, as rats fed the Tucum-do-Cerrado diet had lower muscle glycogen content associated with the upregulation of Slc2a4 and Prkaa1 mRNA levels. Uruno et al. (2016) [46] demonstrated that specific Keap1 knockout mice that abundantly expressed Nrf2 in their skeletal muscle showed reduced blood glucose levels and increased glycogen branching enzyme (GBE) and the phosphorylase b kinase α subunit (Phkα) protein in skeletal muscle compared to wild type mice. They suggested that Nrf2 induction improved glucose tolerance through the increased glucose uptake and utilization, mediated through GBE and Phkα induction and consequently activated muscle glycogen metabolism, resulting in diminished muscle glycogen content. Nrf2 induction also increases energy consumption in the skeletal muscles in specific Keap1 knockout mice that abundantly express Nrf2.

Obesity and insulin resistance are associated with low-grade systemic inflammation and oxidative stress [7]. In agreement with the literature HF diet consumption decreased the enzymatic antioxidant defense, as hepatic GR, GPX, GST, and SOD specific activities were lower than those of the control diet. Although oxidative damage occurred in the liver, carbonyl and MDA concentrations were not affected by HF diet. An et al. (2013) [47] observed a decrease in the polyunsaturated fatty acid to unsaturated fatty acid ratio (PUFA-to-UFA) in the livers of rats treated for 83 days with a HF diet. They suggested that a HF diet enhanced the peroxidation of polyunsaturated fatty acids and thus oxidative stress; under such conditions, apolipoprotein B undergoes proteolysis, which impairs very low-density lipoprotein secretion, contributing to triglyceride accumulation in the liver [48]. In the present study, HF diet treated rats also showed higher hepatic triglyceride concentration in relation to control group rats ($155.80 \pm 21.44$ and $80.81 \pm 23.97$ mg/dL, respectively), therefore, reinforcing that an oxidative stress condition was established and associated with the impairment of enzymatic antioxidant defense.

The consumption of Tucum-do-Cerrado improved the total serum antioxidant capacity independent of diet type, suggesting that the phytochemical compounds found in Tucum-do-Cerrado [49] are bioavailable *in vivo* and display antioxidant activities, such as the reduction or chelation of oxidative metals and scavenging of free radicals. These data also suggest that dietary lipid content does not influence the bioavailability of these phytochemicals, as both the control and HF diet rats showed similar serum FRAP values. Phytochemical compounds also improve endogenous antioxidant defenses by increasing hepatic GR activity in rats fed a HF diet with Tucum-do-Cerrado compared to HF rats and maintain their activity similar to that of the control diet.

A previous study showed that Tucum-do-Cerrado increased nuclear factor erythroid 2-related factor 2 (Nrf2) protein levels, which upregulated antioxidant enzyme expression and attenuated oxidative stress [17]. The authors suggested that this antioxidant response is modulated by phytochemical compounds that induce the expression and activate Nrf2 transcription factor. Nrf2 positively regulates the antioxidant GSH-based system glutathione reductase maintaining intracellular levels of reduced glutathione (GSH) [50].

Nrf2 also mediates the induction of drug-metabolizing enzymes, such as glutathione-S-transferase [50], in agreement with our data, in which Tucum-do-Cerrado consumption increased GST activity in rat liver independent of diet type. Together, these results suggest that Tucum-do-Cerrado consumption promotes antioxidant response in both the control and HF

diet groups, however, in the HF diet group, it reestablished the GSH-based antioxidant system to normal levels.

In the small intestine, the reduction in catalase and glutathione peroxidase activities observed in rats treated with Tucum-do-Cerrado may be attributed to the sparing effect of dietary antioxidants. The high content of polyphenols in Tucum-do-Cerrado increased the exogenous antioxidant concentration in the intestine, leading to an environment with high reducing potential, which decreased the need for enzymatic endogenous antioxidants. Pereira et al. (2014) [51] observed a similar response when rats treated with tropical fruit juice had lower antioxidant enzyme activity than control rats.

According to the literature, phytochemicals also exhibit anti-inflammatory effect by reducing the expression and activation of pro-inflammatory cytokines, such as IL-1β, TNF-α, IL-6 and IL-8 [52]. Unexpectedly, Tucum-do-Cerrado consumption upregulated intestinal Tnf and Il1b mRNA levels and serum IL-6 independent of diet type and serum IL-1b when consumed with a HF diet, suggesting a pro-inflammatory response. Zampelas et al. (2004) [53] when investigating the association between coffee consumption and inflammatory markers, showed that subjects who consumed >200 mL coffee/day had higher serum IL-6, C-reactive protein, TNF-α concentrations compared with coffee nondrinkers. Therefore, the high polyphenol content, more specifically in the intestine than in other tissues, may be related to the greater pro-inflammatory response observed in this tissue.

Considering the high content of phenolic compounds of Tucum-do-Cerrado fruit, we suggest that one or more polyphenols may be responsible by the biological protective effects of Tucum-do-Cerrado observed in the present study. Flavonoids (flavanols, anthocyanins, flavones and flavonols), phenolic acids (hydroxybenzoics and hydroxycinnamics acids) and stilbenes (resveratrol) are three major classes of phenolic compounds already identified in the Tucum-do-Cerrado fruit [14]. Stilbenes seem to be associated with the upregulation of the nuclear factor-erythroid-2-related factor-2 (Nrf2) [16] that improves glucose tolerance, while the flavonoid anthocyanins, responsible for the blue-purple color of many fruits, induces glutathione synthesis and the flavonol, quercetin, acts as a free radical scavenger protecting cells against oxidative stress [54, 55]. Despite flavonoids represent the major content of phenolic compounds in Tucum-do-Cerrado fruit, the biological protective effect of Tucum-do-Cerrado should be related to the type of phytochemical rather than its concentration. An *in vitro* study showed that tucum-do-cerrado peel methanolic extract had the highest phenolic concentration, however the lowest antioxidant activity, compared to the ethanolic extract [14].

## 5. Conclusion

Consumption of Tucum-do-Cerrado may ameliorate impaired glucose utilization in a HF diet-induced obesity model by increasing liver and muscle glucose uptake and oxidation. These data suggest that its' consumption improves muscle glucose oxidation in non-obese and obese rats. This response may be related to the improvement in the total antioxidant capacity of rats.

## Acknowledgments

We would like to thank Mr. Luiz Eduardo da Silva Araújo for technical assistance, as well as Editage (www.editage.com) for English language editing.

## Author Contributions

**Conceptualization:** Ananda de Mesquita Araújo, Sandra Fernandes Arruda.

**Data curation:** Ananda de Mesquita Araújo, Sandra Fernandes Arruda.

**Formal analysis:** Ananda de Mesquita Araújo, Sandra Fernandes Arruda.

**Funding acquisition:** Ananda de Mesquita Araújo, Sandra Fernandes Arruda.

**Investigation:** Ananda de Mesquita Araújo, Sandra Fernandes Arruda.

**Methodology:** Ananda de Mesquita Araújo, Sandra Fernandes Arruda.

**Project administration:** Sandra Fernandes Arruda.

**Resources:** Sandra Fernandes Arruda.

**Software:** Ananda de Mesquita Araújo, Sandra Fernandes Arruda.

**Supervision:** Ananda de Mesquita Araújo, Sandra Fernandes Arruda.

**Validation:** Ananda de Mesquita Araújo, Sandra Fernandes Arruda.

**Visualization:** Ananda de Mesquita Araújo, Sandra Fernandes Arruda.

**Writing – original draft:** Ananda de Mesquita Araújo, Sandra Fernandes Arruda.

**Writing – review & editing:** Ananda de Mesquita Araújo, Sandra Fernandes Arruda.

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
