## [Decision Letter · Decision Letter 0]

14 Aug 2023

PONE-D-23-20187Ameliorating the impairment of glucose utilization in a high-fat diet-induced obesity model through the consumption of Tucum-do-Cerrado (Bactris Setosa Mart.)PLOS ONE

Dear Dr. Araújo,

Thank you for submitting your manuscript to PLOS ONE. After careful consideration, we feel that it has merit but does not fully meet PLOS ONE’s publication criteria as it currently stands. Therefore, we invite you to submit a revised version of the manuscript that addresses the points raised during the review process.

We look forward to receiving your revised manuscript.

Kind regards,

Anand Thirupathi

Academic Editor

PLOS ONE

Reviewers' comments:

Reviewer's Responses to Questions

**Comments to the Author**

1. Is the manuscript technically sound, and do the data support the conclusions?

Reviewer #1: Yes

Reviewer #2: Yes

Reviewer #3: Yes

2. Has the statistical analysis been performed appropriately and rigorously? 

Reviewer #1: Yes

Reviewer #2: N/A

Reviewer #3: Yes

3. Have the authors made all data underlying the findings in their manuscript fully available?

Reviewer #1: Yes

Reviewer #2: Yes

Reviewer #3: Yes

4. Is the manuscript presented in an intelligible fashion and written in standard English?

Reviewer #1: Yes

Reviewer #2: Yes

Reviewer #3: Yes

5. Review Comments to the Author

Reviewer #1: Good efforts by the authors.

Q1. Fig.1 is not called out in the results section. Since the body weight of rats are overlapping, a line graph would be more appropriate for better visualization. This should be considered.

Q2. The author should provide the 95% confidence interval for all the p-values. Since the SEM has been used for plotting graphs, there are variations among groups which overlap with other groups, yet significantly different. These data are hidden and opaque to readers.

Q3. Labelling of individual graphs of each figure and appropriate callouts should be done to enhance the readers' attention.

Q4. Fig. 4,6, and 7. What does the word "mRNAr" indicates?

Q5. Authors did not provide the primers for TNFa and IL1b.

Q6. Fig2 A- GST activity significance value of HF/TUC is suspicious. This puts the authors conclusion "This response may be related to the improvement in the total antioxidant capacity of rats" in question. Please clarify the statistics

Suggestion to author to refer for graphs https://clauswilke.com/dataviz/time-series.html

Reviewer #2: High-fat diet fed rats were used in the manuscript, and the glucose utilization was impaired, which could be ameliorated with consumption of Tucum-do-Cerrado, investigation of the underlying mechanisms revealed the increased liver and muscle glucose uptake and oxidation. However, there are some issues to be done to improve the manuscript.

1. The powder of Tucum-do-Cerrado extraction was used for the diet supplements, and could ameliorate the glucose metabolism disorders induced by high-fat diet in the manuscript, and it can also protect tissues against iron-induced oxidative stress in the published work. It is wondering what main component of Tucum-do-Cerrado is protective, which could be discussed or referenced.

2. Although the freezing-dried Tucum-do-Cerrado extraction could induce the changes of muscle glycogen, it has no effects on fasting plasma glucose, body weight and insulin level, which is contrary to the reports in 2020 (doi: 10.1039/D0FO01912G.), is it due to the different dosage? What is the relative dose of the extraction for human beings?

3. In the section of Results, there was no description of Figure 1, which should be inserted in corresponding position.

4. In table 2 and 3, there are labels of a, b or c in graphs, which means the p value? It is recommended to add the information in figure legends respectively.

5. In table 3, check the position of ±, if there is no error for its position, what it represents should be listed in figure legends. Moreover, there are significant changes according to p value < 0.05 such as 0.021 or 0.018, but the description in the main text is no alteration, which should be checked and corrected.

6. In figure 4a, 6a and 7b, what does “r” in “mRNAr” mean? The spelling of B-actin should be β-actin.

7. In the top left corner of the column in figures, there are illustrations of diet, Tucum, diet x Tucum with data, which should also be explained in figure legends.

8. Confirm it is glucosidade or glucosidase in the subtitle of 2.8.1?

9. For statistics, two-way ANOVA is preferable to obtain p value.

Reviewer #3: Dear writers, the article is very well written, it was possible to make a practical and very precise reading, coherently relating the results and discussing them with precision. Like all scientific work, we know that some errors end up passing through and we don't realize it, so I'll make some remarks about the article here.

I noticed a few times in the articles that some acronyms or symbols are described differently. Ex: 98°C - 4oC. It would be of great interest to review the acronyms and symbols throughout the article in order to create a single standard for the presentation of the work.

In addition, some spacing after punctuations and/or symbols are different in some moments of the article (Ex: Fig. 6A - Fig. 5 A). Please, review this too.

in topic 2.2 treatment; the number of animals is spelled out, in which case it is correct to use "24".

The title of item 2.8.1 is misspelled, "Intestinal α-glucosity" please correct for α-glucosidase.

Finally, something that caught my attention was the significant differences (Ex: p<0.0001), it is not common to use the point to separate the integer part from the decimal part, this is only common in English-speaking countries. Therefore, the correct thing is to review all the significant differences represented in the article and replace the points with commas.

6. PLOS authors have the option to publish the peer review history of their article (what does this mean?). If published, this will include your full peer review and any attached files.

Reviewer #1: No

Reviewer #2: No

Reviewer #3: No

---

## [Author Response · Author response to Decision Letter 0]

22 Sep 2023

To: Anand Thirupathi 

 Academic Editor 

Subject: Revised version of the manuscript [PONE-D-23-20187] – [EMID:49af62f45a6b65c6]

Dear Anand Thirupathi

Thank you by the opportunity to submit a revised version of our manuscript. Below are the responses to each point brought up by the reviewers #1, #2 and #3. Some changes made were marked in the revised manuscript copy and the answers to some reviewer’s questions are presented below. 

Yours sincerely,

Ananda de Mesquita Araújo. Faculty of Health Sciences, Campus Universitário Darcy Ribeiro, Universidade de Brasília, Brasília, Distrito Federal, 70910-900, Brazil. Phone: +55 (61) 31071635. e-mail: nutriananda.araujo@gmail.com

Reviewer #1: 

Q1. Fig.1 is not called out in the results section. Since the body weight of rats are overlapping, a line graph would be more appropriate for better visualization. This should be considered.

The requested information was inserted in the manuscript in the section results.

Data were presented in a line graph.

Q2. The author should provide the 95% confidence interval for all the p-values. Since the SEM has been used for plotting graphs, there are variations among groups which overlap with other groups, yet significantly different. These data are hidden and opaque to readers.

All data in tables and figures are now expressed as means and 95% confidence interval.

Q3. Labelling of individual graphs of each figure and appropriate callouts should be done to enhance the readers' attention.

We have addressed the request for labeling of the figures and the inclusion of appropriate callouts. All identifications have been done as per your suggestion and are presented in the revised manuscript file to enhance readers' comprehension.

Q4. Fig. 4,6, and 7. What does the word "mRNAr" indicates?

It was a mistake. The correct acronym is mRNA. It was corrected in all figures.

Q5. Authors did not provide the primers for Tnfa and Il1b.

The primers sequences for Tnfa and Il1b were included in table 1. 

Q6. Fig2 A- GST activity significance value of HF/TUC is suspicious. This puts the authors conclusion "This response may be related to the improvement in the total antioxidant capacity of rats" in question. Please clarify the statistics

The presentation format of data in tables and figures has been modified to improve reader understanding. In Fig 2A-GST a significant difference was obtained for diet, where HF diet decreased GST activity compared to CT diet [634.26 (583.60 - 684.93) vs 762.45 (711.78 - 813.11)]. A significant difference was also obtained for Tucum-do-Cerrado consumption, where Tucum-do-Cerrado increased GST activity independent of diet type [746.81 (696.15-797.48) vs 649.90 (599.23-700.56)]. We believe that the format used to present the data led to a misinterpretation of the results, therefore we have modified the presentation format of data to make it clearer. Below is the two-way ANOVA statistical analysis of Fig 2A-GST. 

Reviewer #2: High-fat diet fed rats were used in the manuscript, and the glucose utilization was impaired, which could be ameliorated with consumption of Tucum-do-Cerrado, investigation of the underlying mechanisms revealed the increased liver and muscle glucose uptake and oxidation. However, there are some issues to be done to improve the manuscript.

1. The powder of Tucum-do-Cerrado extraction was used for the diet supplements, and could ameliorate the glucose metabolism disorders induced by high-fat diet in the manuscript, and it can also protect tissues against iron-induced oxidative stress in the published work. It is wondering what main component of Tucum-do-Cerrado is protective, which could be discussed or referenced.

The main components of Tucum-do-Cerrado that may exert the protective effect was discussed and referenced in the discussion section.

2. Although the freezing-dried Tucum-do-Cerrado extraction could induce the changes of muscle glycogen, it has no effects on fasting plasma glucose, body weight and insulin level, which is contrary to the reports in 2020 (doi: 10.1039/D0FO01912G.), is it due to the different dosage? What is the relative dose of the extraction for human beings?

Three main factors may explain the contradictory reports in 2020 (doi: 10.1039/D0FO01912G.):

The discrepancies between the study results could be attributed to different treatment protocols.

First, the reports in 2020 used an ethanol/water extract, the extracts were concentrated using a vacuum rotatory evaporator. Extraction improves the bioavailability of polyphenols and also increases their concentration in relation to fruit matrix in natura. Moreover, the reports in 2020 purified the extract using a resin that reduce free sugars, pectins, and other impurities, which improves polyphenols bioavailability.

Second, the animal model used by the reports in 2020 is C57BL/6J mice, while we used Wistar rats. An underlying genetic predisposition to be obesity prone or resistant is also shown in animal models. Rats and mice known as the standard models for studying dietary obesity are different in their susceptibility to obesity: outbred Sprague–Dawley rats, Wistar rats and C57BL/6C mice can be easily categorized to prone and resistant phenotypes with ad libitum access to high-fat diets (Hariri 2010). Hariri N, Thibault L. High-fat diet-induced obesity in animal models. Nutr Res Rev. 2010 Dec;23(2):270-99. doi: 10.1017/S0954422410000168. 

Third, the reports in 2020 administrated the extract by gavage, which may improve polyphenols bioavailability.

3. In the section of Results, there was no description of Figure 1, which should be inserted in corresponding position.

The requested information was inserted in the manuscript in the section results.

4. In table 2 and 3, there are labels of a, b or c in graphs, which means the p value? It is recommended to add the information in figure legends respectively.

The information about the meanings of p values has been added in the legends of tables and figures. Also, the presentation format of data in tables and figures has been modified to improve reader understanding.

5. In table 3, check the position of ±, if there is no error for its position, what it represents should be listed in figure legends. Moreover, there are significant changes according to p value < 0.05 such as 0.021 or 0.018, but the description in the main text is no alteration, which should be checked and corrected.

It was an error of formatting, and therefore corrected in this new version of the manuscript. 

In the results section in the subtitle 3.1. the text describes that the carbonyl concentration in liver (p = 0.021) and MDA (p = 0.018) in the intestine showed significant changes. No significant alteration was described for hepatic MDA and intestinal carbonyl, in accordance with data presented in Table 3. 

In the discussion section (ninth paragraph) the description “Although oxidative damage occurred in the liver, carbonyl and MDA concentrations were not affected” refers to the high-fat diet compared to control diet. Hepatic carbonyl (p = 0.021) and MDA in the intestine (p = 0.018) were altered by Tucum-do-Cerrado treatment. Therefore, to make the description clearer, it was included at the end of the sentence “by high-fat diet”. 

6. In figure 4a, 6a and 7b, what does “r” in “mRNAr” mean? The spelling of B-actin should be β-actin.

It was a mistake. The correct acronym is mRNA. It was corrected in all figures.

The spelling of β-actin has been corrected throughout the text, and in figures.

7. In the top left corner of the column in figures, there are illustrations of diet, Tucum, diet x Tucum with data, which should also be explained in figure legends.

The meaning of diet, Tucum, and diet x Tucum has been explained in the figure legends.

It was also included the p acronym in all figures.

8. Confirm it is glucosidade or glucosidase in the subtitle of 2.8.1?

The correct spelling is glucosidase. It was corrected in the text.

9. For statistics, two-way ANOVA is preferable to obtain p value.

The statistical analyzes were performed using two-way ANOVA. Maybe the description presented in 2.11 item is confusing, therefore we rewrote the paragraphs to make it clearer.

Reviewer #3: 

I noticed a few times in the articles that some acronyms or symbols are described differently. Ex: 98°C - 4oC. It would be of great interest to review the acronyms and symbols throughout the article in order to create a single standard for the presentation of the work.

The acronyms and symbols were revised throughout the article.

In addition, some spacing after punctuations and/or symbols are different in some moments of the article (Ex: Fig. 6A - Fig. 5 A). Please, review this too.

The spacing after punctuations and/or symbols were revised throughout the article.

in topic 2.2 treatment; the number of animals is spelled out, in which case it is correct to use "24".

It was correct in the text.

The title of item 2.8.1 is misspelled, "Intestinal α-glucosity" please correct for α-glucosidase.

It was correct in the text.

Finally, something that caught my attention was the significant differences (Ex: p<0.0001), it is not common to use the point to separate the integer part from the decimal part, this is only common in English-speaking countries. Therefore, the correct thing is to review all the significant differences represented in the article and replace the points with commas.

According to the submission guidelines the manuscripts must be submitted in English. Therefore, we used the point to separate the integer part from the decimal part.

---

## [Decision Letter · Decision Letter 1]

17 Oct 2023

Ameliorating the impairment of glucose utilization in a high-fat diet-induced obesity model through the consumption of Tucum-do-Cerrado (Bactris Setosa Mart.)

PONE-D-23-20187R1

Dear Dr. Araújo,

We’re pleased to inform you that your manuscript has been judged scientifically suitable for publication and will be formally accepted for publication once it meets all outstanding technical requirements.

Kind regards,

Anand Thirupathi

Academic Editor

PLOS ONE

Additional Editor Comments (optional):

Reviewers' comments:

Reviewer's Responses to Questions

**Comments to the Author**

1. If the authors have adequately addressed your comments raised in a previous round of review and you feel that this manuscript is now acceptable for publication, you may indicate that here to bypass the “Comments to the Author” section, enter your conflict of interest statement in the “Confidential to Editor” section, and submit your "Accept" recommendation.

Reviewer #1: All comments have been addressed

Reviewer #2: All comments have been addressed

2. Is the manuscript technically sound, and do the data support the conclusions?

Reviewer #1: Yes

Reviewer #2: Yes

3. Has the statistical analysis been performed appropriately and rigorously? 

Reviewer #1: Yes

Reviewer #2: Yes

4. Have the authors made all data underlying the findings in their manuscript fully available?

Reviewer #1: Yes

Reviewer #2: Yes

5. Is the manuscript presented in an intelligible fashion and written in standard English?

Reviewer #1: Yes

Reviewer #2: Yes

6. Review Comments to the Author

Reviewer #1: All the previous questions were addressed.

I appreciate the great efforts taken to visualize the data in graphs.

The legends of graph in individual figures (i.e., a, b, c, etc., of figures) are not consistent throughout the manuscript.

Do arrange the legends.

Reviewer #2: The revised manuscript has corrected the errors in the tables and figures and results descriptions, and the statistics methods have been used as required. Moreover, the main components of Tucum-do-Cerrado that may exert the protective effect have also been discussed at the end of the revised manuscript. Therefore, the present form of manuscript has been been improved a lot, which could be considered for acception.

7. PLOS authors have the option to publish the peer review history of their article (what does this mean?). If published, this will include your full peer review and any attached files.

Reviewer #1: **Yes: **Kannathasan Thetchinamoorthy

Reviewer #2: No

---

## [Editor Report · Acceptance letter]

26 Oct 2023

PONE-D-23-20187R1 

Ameliorating the impairment of glucose utilization in a high-fat diet-induced obesity model through the consumption of Tucum-do-Cerrado (*Bactris Setosa Mart.*) 

Dear Dr. Araújo:

I'm pleased to inform you that your manuscript has been deemed suitable for publication in PLOS ONE. Congratulations! Your manuscript is now with our production department. 

Kind regards, 

on behalf of

Dr. Anand Thirupathi 

Academic Editor

PLOS ONE